# Developing an Atlas of Harmful Algal Blooms in the Red Sea: Linkages to Local Aquaculture

**Elamurugu Alias Gokul [1], Dionysios E. Raitsos [2] , John A. Gittings [1] and Ibrahim Hoteit [1],***

[1] Earth Science and Engineering (ErSE), King Abdullah University of Science and Technology (KAUST), Thuwal 23955, Saudi Arabia; rajadurai.elamurugu@kaust.edu.sa (E.A.G.); john.gittings@kaust.edu.sa (J.A.G.)

[2] Department of Biology, National and Kapodistrian University of Athens (NKUA), 15784 Athens, Greece; draitsos@biol.uoa.gr

* Correspondence: Ibrahim.hoteit@kaust.edu.sa

**Abstract:** Harmful algal blooms (HABs) are one of the leading causes of biodiversity loss and alterations to ecosystem services. The Red Sea is one of the least studied large marine ecosystems (LMEs), and knowledge on the large-scale spatiotemporal distribution of HABs remains limited. We implemented the recently developed remote sensing algorithm of Gokul et al. (2019) to produce a high-resolution atlas of HAB events in the Red Sea and investigated their spatiotemporal variability between 2003 and 2017. The atlas revealed that (i) the southern part of the Red Sea is subject to a higher occurrence of HABs, as well as long-lasting and large-scale events, in comparison to the northern part of the basin, and (ii) the Red Sea HABs exhibited a notable seasonality, with most events occurring during summer. We further investigated the potential interactions between identified HAB events and the National Aquaculture Group (NAQUA), Al-Lith (Saudi Arabia)—the largest aquaculture facility on the Red Sea coast. The results suggest that the spatial coverage of HABs and the elevated chlorophyll-a concentration (*Chl-a*) ($> 1$ mg m$^{-3}$; a proxy for high nutrient concentration), in the coastal waters of Al-Lith during summer, increased concurrently with the local aquaculture annual production over a nine-year period (2002–2010). This could be attributed to excessive nutrient loading from the NAQUA facility's outfall, which enables the proliferation of HABs in an otherwise oligotrophic region during summer. Aquaculture is an expanding, high-value industry in the Kingdom of Saudi Arabia. Thus, a wastewater management plan should ideally be implemented at a national level, in order to prevent excessive nutrient loading. Our results may assist policy-makers' efforts to ensure the sustainable development of the Red Sea's coastal economic zone.

**Keywords:** harmful algal blooms; Red Sea; remote sensing; aquaculture

## 1. Introduction

In global marine ecosystems, harmful algal blooms (HABs) are known for the rapid increase of algal biomass and occasionally the production of toxins [1]. HABs are often associated with a broad range of ecological impacts, including increased mortalities of wild and farmed fish, displacement of indigenous species, and alterations to marine food webs, which ultimately have a detrimental effect on human societies [2,3]. HABs constitute a worldwide issue, which is constantly intensified in its frequency and spatial extent, primarily due to oceanic warming (climate change) and increasing anthropogenic pollution [4–7].

The Red Sea, a large marine ecosystem (LME) characterized by thriving coral reef complexes, high levels of biological diversity, and a growing aquaculture industry, is one such region that has been subjected to substantial impacts from HABs [8–17]. A recent study summarized historical HAB events that were previously reported by different field sampling programs in the Red Sea over the last

three decades [18]. The main phytoplankton functional types (PFTs) responsible for the reported HAB events were found to be dinoflagellates, raphidophytes, and cyanobacteria [18–20]. Besides traditional in situ approaches, Gokul et al. [17] developed and validated a remote sensing model to detect and map the spatial distribution of HABs associated with the aforementioned PFTs in the Red Sea.

Several studies focusing on Red Sea HABs have enabled a deeper understanding of their spatiotemporal distribution, species toxicity levels (only possible through in situ measurements) and the underlying oceanographic mechanisms responsible for their outbreaks [12,17,18]. For instance, previous studies have conducted field sampling programs and reported several HAB species including *Kryptoperidinium foliaceum*, *Noctiluca scintillans/miliaris*, *Heterosigma akashiwo*, *Pyrodinium bahamense* var. *bahamense*, *Ostreopsis* sp. and the cyanobacterium *Trichodesmium erythraeum* [8–16]. Mohamed [18] documented the main environmental factors contributing to the dominance of these HAB species. Recently, a remote sensing algorithm was developed to investigate the spatial variability of these HAB species in the Red Sea, in accordance with previously reported events in the Red Sea [17]. However, studies on Red Sea HABs are limited with regards to the investigation of their large-scale spatial distribution, long-term temporal (interannual and/or seasonal) evolution, and interactions with local aquaculture industries. Such information is essential for the regional economy (i.e., fisheries, aquaculture, and tourism) and will support policy-makers to optimize the sustainable management of the Red Sea's coastal economic zone. For instance, the National Aquaculture Group (NAQUA) located at the coastal city Al-Lith is the largest aquaculture facility in the Red Sea and is an important source of food and economic prosperity to the Kingdom of Saudi Arabia. Thus, routine monitoring of HABs over the relevant spatial–temporal scales is required in order to develop a more sustainable aquaculture operation and to improve coastal management strategies [11,20,21]. Here, we implemented the remote sensing algorithm developed by Gokul et al. [17] on a 15-year satellite-derived remote sensing reflectance ($R_{rs}$) dataset (2003–2017) to produce an atlas of HAB events in the Red Sea. Furthermore, we investigated the interactions between HAB events and Al-Lith's NAQUA aquaculture facility. Our HAB Red Sea atlas will assist the policy-makers with regional decision-making and risk mitigation processes that are associated with environmental and aquaculture management.

## 2. Materials and Methods

This study used the recently developed remote sensing algorithm of Gokul et al. [17] to develop an atlas of HABs in the Red Sea and then investigate their possible linkages to the largest aquaculture of the basin. This remote sensing algorithm was developed by applying high resolution satellite-derived $R_{rs}$ observations and available in situ datasets to a second order derivative technique for detecting the presence/absence and mapping the spatial distribution of HABs in the Red Sea [17].

### 2.1. Satellite Datasets

Satellite remote sensing datasets of $R_{rs}$ were acquired from the Moderate Resolution Imaging Spectroradiometer (MODIS) and the Medium Resolution Imaging Spectrometer (MERIS). Satellite-derived observations of Chlorophyll-a concentration (*Chl-a*) were obtained from the Ocean Colour Climate Change Initiative (OC-CCI) of the European Space Agency (ESA). We obtained MODIS-Aqua and MERIS datasets from the NASA ocean color archive (https://oceancolor.gsfc.nasa.gov), whereas the OC-CCI dataset is available at http://www.esa-oceancolour-cci.org.

First, MODIS-Aqua Level-3 mapped data of $R_{rs}$ (for wavelengths 412, 443, 488, 531, 547, 667, and 678 nm) were acquired at a spatial resolution of 4 km, and an 8-day temporal resolution, for the Red Sea, covering a 15-year period spanning from 2003 to 2017. Secondly, we used Level-2 MERIS-derived $R_{rs}$ data (for wavelengths 413, 443, 490, 510, 560, 620, 665, 681, and 709 nm) at a 1.2 km and 1-day resolution for a 9-year period (2002–2010), covering the region of Al-Lith, Saudi Arabia. The region of Al-Lith and its aquaculture facility, is located between 19°12′0″N and 20°6′0″N latitude and 40°0′0″E and 42°0′0″E longitude (see red box in Figure 1). Previous studies have suggested that the aforementioned region is subject to the substantial nutrient loading from NAQUA aquaculture

effluents [11,20,21]. Finally, version 4.1 of the ESA OC-CCI product was used. This product is comprised of merged and bias-corrected *Chl-a* data from MERIS, MODIS, Sea-Viewing Wide Field-of-View Sensor (SeaWiFS), and the Visible Infrared Imaging Radiometer Suite (VIIRS) satellite sensors. Level-3 mapped data of *Chl-a* were obtained at 4 km spatial resolution and 8-day temporal resolution for the period 2002 to 2010 in the Al-Lith region. We note that standard remote sensing algorithms may systematically overestimate *Chl-a* concentrations when compared with in situ measurements in the Red Sea. This is mainly due to excess chromophoric dissolved organic matter (CDOM) absorption per unit chlorophyll in this basin when compared with average global conditions [22]. However, Brewin et al. [23] and Racault et al. [24] have demonstrated that the OC-CCI derived *Chl-a* dataset exhibits a good agreement with independent in situ *Chl-a* datasets in coastal and open waters of the Red Sea. In addition, several studies have suggested that the OC-CCI product is characterized by higher data availability in comparison to single-sensor-based missions [24–26]. Thus, we are confident in using the OC-CCI derived *Chl-a* dataset for investigating interactions between HABs and local aquaculture in the coastal waters of Al-Lith.

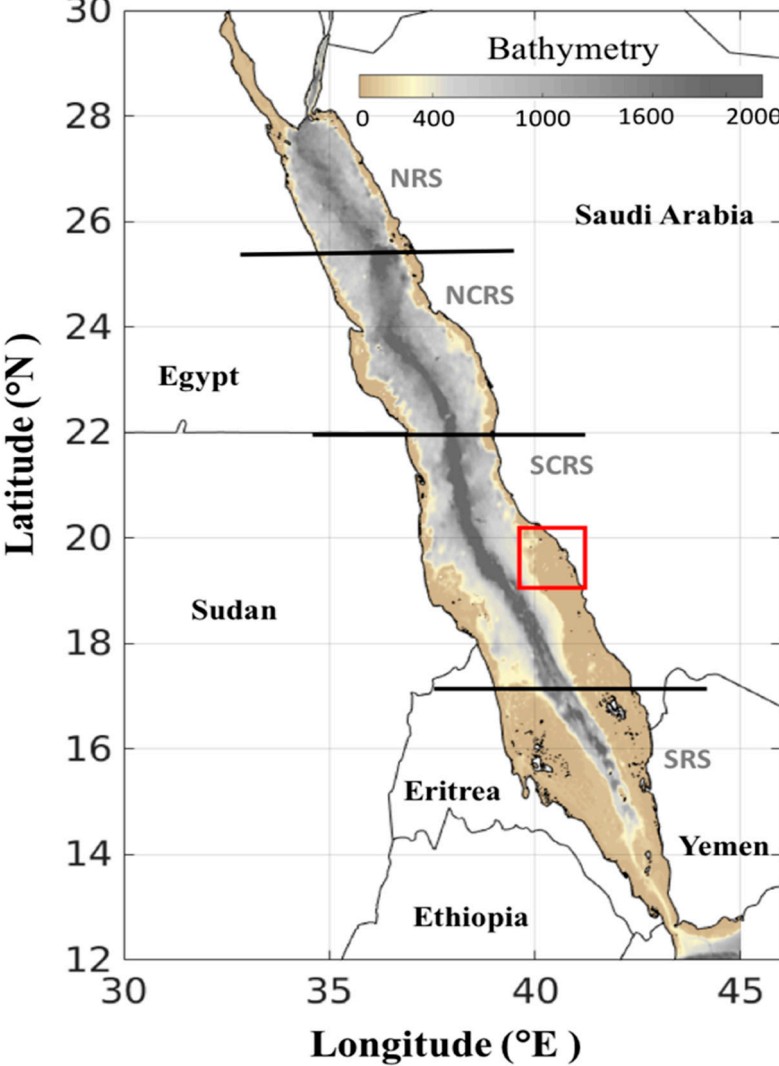

**Figure 1.** Map showing the bathymetry of the Red Sea (acquired from the General Bathymetric Chart of the Oceans (GEBCO_2014 Grid, version 20150318, http://www.gebco.net/)). The Red Sea is divided into four different provinces, following Raitsos et al. (2013): the Northern Red Sea (NRS), North-Central Red Sea (NCRS), South-Central Red Sea (SCRS) and Southern Red Sea (SRS). We also note that the red box indicates the geographical location of the Al-Lith region, Saudi Arabia.

### 2.2. Aquaculture Production Dataset

Aquaculture production data were acquired from the Sea Around Us project (http://www.seaaroundus.org) for examining the possible linkages between aquaculture and HABs. The Sea Around Us project is based on a global analysis that aims to generate a database of the annual reported aquaculture production in all maritime countries from 1950 to 2010 [27–29]. The project has also documented the annual mariculture production for countries bordering the Red Sea, including Egypt, Sudan, Eritrea, Yemen, Saudi Arabia, Jordan, and Israel. The average annual production of NAQUA, the largest shrimp farming aquaculture facility in the Red Sea, was reported by this project for the period spanning 1985 to 2010 [27–29]. Here, we utilized the average annual production dataset of the NAQUA aquaculture facility for the period 2002 to 2010, corresponding with the available MERIS derived $R_{rs}$ observations. We note that the aquaculture production dataset was obtained from a report submitted by the Kingdom of Saudi Arabia to the Food and Agricultural Organization (FAO) of the United Nations (http://www.fao.org/fishery/countrysector/naso_saudiarabia/en).

### 2.3. Approach

Our approach is comprised of two parts: (1) Produce an atlas of HAB events in the Red Sea, and (2) Investigate the interactions between HABs and aquaculture. For the first aim, we implemented the remote sensing algorithm developed recently by Gokul et al. [17] on a 15-year MODIS-derived $R_{rs}$ dataset (2003–2017) to reveal spatial patterns in HAB frequency, duration, and spatial coverage. We also evaluated the temporal patterns of HABs in terms of the total number of events in the Red Sea over three main seasons (winter (January to April), summer (May to August), and autumn (September to December). In order to provide a more comprehensive spatial and temporal description of HAB events, we divided the Red Sea into four regions following Raitsos et al. [30]: the Northern Red Sea (NRS; 25.5–30° N), the North-Central Red Sea (NCRS; 22–25.5° N), the South-Central Red Sea (SCRS; 17.5–22° N), and the Southern Red Sea (SRS; 12.5–17.5° N) (Figure 1). For the second part of our analysis, we applied the algorithm to MERIS-derived $R_{rs}$ in the Al-Lith region over a 9-year period (2002–2010), in order to examine the possible links between HAB events and the largest aquaculture facility in the Red Sea. In addition, satellite-derived *Chl-a* concentrations (OC-CCI) were analysed in the coastal waters of Al-Lith over the period 2002 to 2010, in order to study the potential links between the aquaculture facility and HABs.

## 3. Results

### 3.1. Atlas of HABs in the Red Sea

We produced a high-resolution atlas of HAB events to depict their overall spatial variability in the Red Sea. Throughout the 15-year period (2003–2017), the eastern coast of the Red Sea (excluding the NRS) hosts the highest number of HAB events, whilst more persistent events can be observed in a substantial part of the SRS (Figure 2a,b). Large-scale events were apparent along the western coast and open waters of the SCRS and the SRS (Figure 2c). Additionally, Red Sea HABs exhibit a notable seasonality in terms of the total number of events, with most HABs occurring during summer (at least 30 events), and fewer events occurring during winter and autumn (at least 21 events) (Figure 3). In general, we observed the highest number of HAB events in the SRS, followed by the SCRS and then the NCRS (Figure 2). HAB events in the Red Sea appear to decrease northward and no events were detected by the remote sensing algorithm in the NRS over the study period (Figure 2). Next, we provide spatial and seasonal descriptions of HAB events for the three Red Sea provinces where HABs were detected.

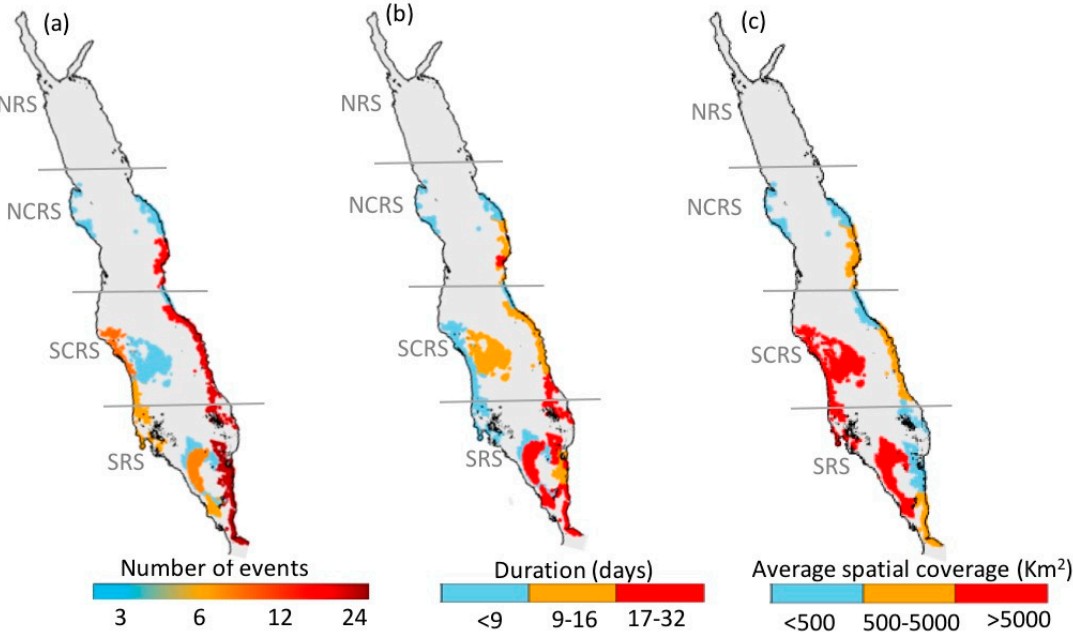

**Figure 2.** Atlas of harmful algal blooms (HAB) events over the Red Sea during 2003–2017. The maps represent: (**a**) the total number of HAB events, (**b**) the average duration of HAB events, and (**c**) the average spatial coverage of HAB events.

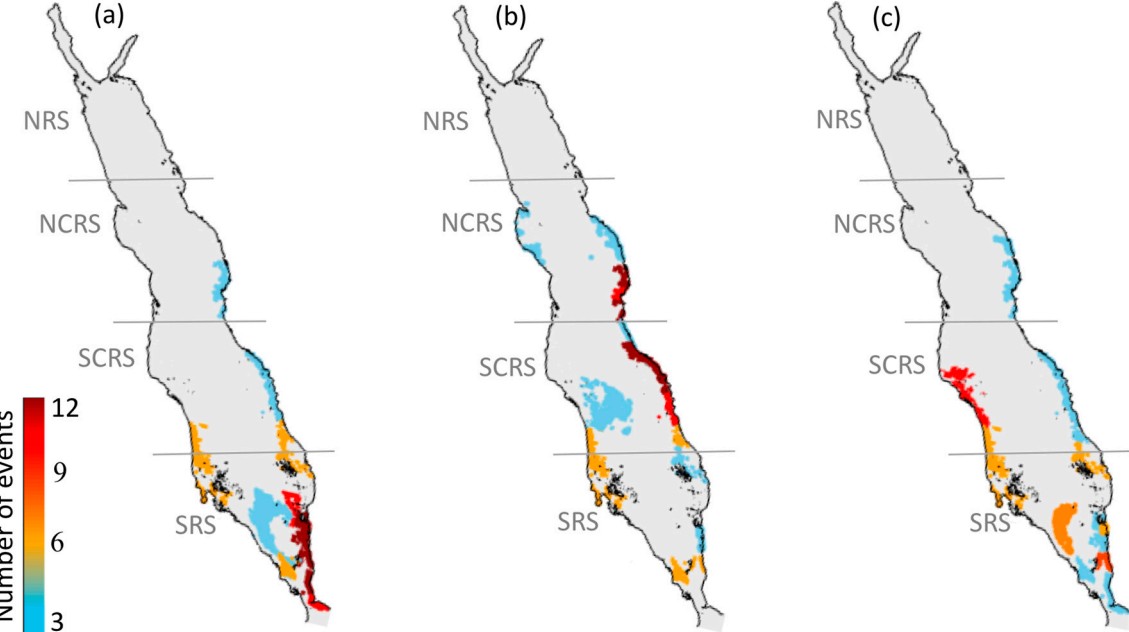

**Figure 3.** Seasonal occurrence of the number of HAB events for the period 2003–2017. The maps represent: (**a**) Winter climatology (January to April), (**b**) Summer climatology (May–August), and (**c**) Autumn climatology (September–December).

### 3.1.1. North-Central Red Sea

The western coast of the NCRS experienced a small number of HAB events over the study period (at least three events, Figure 2a), which were generally of short duration (less than nine days, Figure 2b) and low spatial coverage (less than 500 km$^2$, Figure 2c). Most of the east coast experienced fewer events, apart from the southern part of the NCRS, where the highest number of events (at least 16) were recorded. HABs along the east coast exhibit a longer duration in comparison to the west coast,

ranging from 9 to 16 days and a mean spatial coverage ranging from 500 to 5000 km$^2$. The seasonality of HAB events was investigated in the NCRS and is presented in Figure 3a–c. Overall, we observed the highest number of HAB events (at least 12 events) during summer and fewer events during winter and autumn (at least 3 events). Over the 15-year period, no HAB events were detected on the western coastline of the NCRS region during winter and autumn, while a small number of events were observed during summer (at least three events). On the eastern coast, the total number of HAB events was at least 12 in the southern part of NCRS and decreased towards the northern part of the region to at least 3 events during summer (Figure 3b).

### 3.1.2. South-Central Red Sea

In the SCRS, the majority of the HAB events were detected over the eastern coastline, while fewer events were observed along the western coast and open waters of the region (Figure 2a). The duration of HAB events appeared to be considerably higher along the eastern coastline and open waters (9–16 days), in comparison to the western coast, which exhibited shorter events (less than 9 days) (Figure 2b). On the east coastline, the mean spatial coverage is less than 500 km$^2$ in the northern part of SCRS and increases towards the southern part of the region at least ~5000 km$^2$. Noticeably higher spatial coverage was observed over the western coast and the open waters of the region (>5000 km$^2$) (Figure 2c). Overall, the total number of HABs was highest during summer in the SCRS region (at least 12 events), followed by autumn (at least 9 events) and a moderately low number of events during winter (at least 6 events) (Figure 3a–c). Throughout the 15-year study, a small number of events (at least three events) were detected in the open waters of SCRS during summer, whilst no HAB events were observed in the open waters during winter and autumn (Figure 3a–c). The eastern coastline of the SCRS experienced few events during the winter and autumn (at least 6 events), while maximum HAB events was observed during summer (at least 12 events). In contrast, we observed the largest number of HAB events along the western coast of SCRS during autumn (at least nine events) while relatively fewer events over winter and summer seasons (at least six events).

### 3.1.3. Southern Red Sea

The SRS exhibited a higher occurrence of HABs than anywhere else in the Red Sea (reaching at least 24 events), which were characterized by a longer duration and higher spatial coverage. The eastern coastline of the SRS region experienced the highest number of HAB events (at least 18), while fewer events occurred on the west coast and the open waters of the region (Figure 2a). A substantial part of the SRS was observed to have more long-lasting events (17–32 days, Figure 2b). The mean spatial coverage on the east coast was mostly low (500 km$^2$ or less), whilst the southern part of the SRS had spatial coverage ranging from 500–5000 km$^2$. The mean spatial coverage of HAB events reached values as high as ~42,904 km$^2$ in the open waters of the SRS (Figure 2c). The seasonal investigation of the SRS revealed that the region is subjected to fewer events during summer (at least six events). The total number of HAB events in the SRS was higher during winter (at least 12 events) than autumn (at least 9 events) (Figure 3a–c).

Overall, our atlas reveals that regions along the east coast of the Red Sea (excluding the NRS) appear to be areas of higher risk of occurrence of HABs. The SCRS and SRS exhibit the highest numbers of HABs on the eastern coastline (reaching at least 18 events), as well as persistent and fairly widespread events (at least ~5000 km$^2$). It is worthwhile to note that most of the HAB events on the east coast of the SCRS region were observed during summer (Figure 3b). In the following Section, we utilize the available MERIS derived $R_{rs}$ observations (2002–2010) to examine the possible linkages between HABs in the SCRS and the local aquaculture facility.

### 3.2. Interactions of HABs with Aquaculture

In this Section, we investigated the possible interactions between HABs in the Al-Lith and the nearby NAQUA aquaculture facility between 2002–2010 (see Data and methods). First, we compared

the spatial coverage of HAB events (in km$^2$) in the coastal waters of Al-Lith during the summer season and the average annual production (in tonnes) from the shrimp farming aquaculture facility (Figure 4b). In general, the spatial coverage of HAB events in the Al-Lith region during summer was found to increase in conjunction with the local aquaculture production over the period 2002–2010 (Figure 4b). Before 2005, the spatial coverage of HABs was low (on average 65 km$^2$) near the aquaculture site over the summer season, coinciding with fairly low levels of annual aquaculture production (~4000 tonnes). Following 2005, the spatial coverage with the presence of HABs rapidly increased and reached a maximum of 392 km$^2$ by 2010, corresponding with the highest production of local aquaculture (22,370 tonnes year$^{-1}$). In order to further examine the presence of HABs in Al-Lith coastal waters over the summer season, we presented a seasonal time series of satellite-derived *Chl-a* concentrations (as a proxy of nutrient concentrations) over three different seasons in the coastal waters of Al-Lith during 2002–2010 (see Data and Methods). During summer, *Chl-a* concentrations were higher (>1 mg m$^{-3}$), in comparison to winter and autumn over the nine-year period. In addition, we noticed that summer *Chl-a* concentrations exhibit a consistent increase over the period of 2002–2010 in the coastal waters of Al-Lith (Figure 4c).

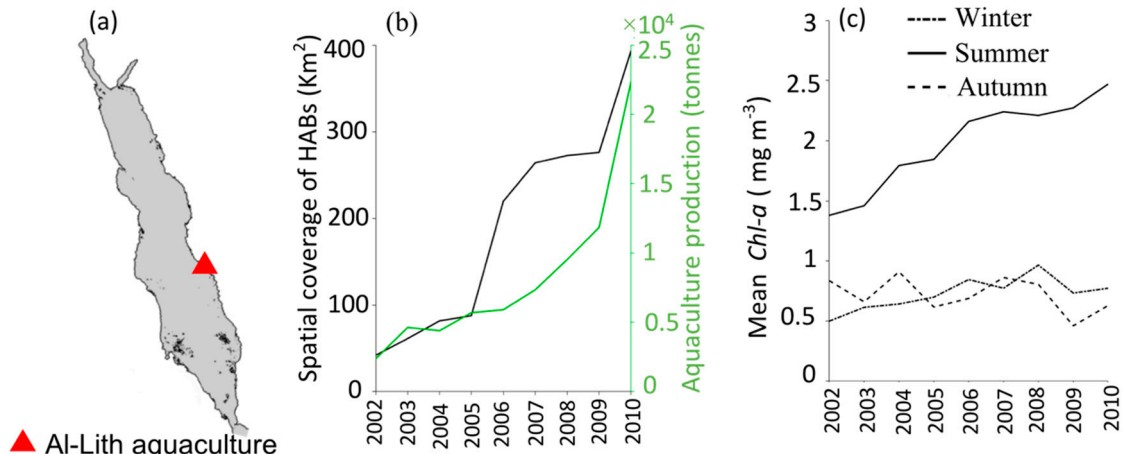

**Figure 4.** (**a**) Map showing the location of aquaculture facility in Al-Lith along the east coast of the Red Sea. (**b**) Relationship between the spatial coverage of MERIS-derived HABs (in km$^2$) in the Al-Lith region and the average annual production (in tonnes) of Al-Lith's aquaculture facility over the period between 2002 and 2010. (**c**) Seasonal time series of satellite-derived *Chl-a* in the coastal waters of Al-Lith over the period 2002 to 2010. The *Chl-a* dataset was acquired from version 4.1 of the ESA OC-CCI product and the annual aquaculture production dataset was obtained from the Sea Around Us project (http://www.seaaroundus.org/).

## 4. Discussion

Over the past few decades, HABs have adversely affected coastal ecosystem services in the Red Sea, including fisheries, aquaculture, and tourism [8–17]. However, prior to this study, research dedicated to the investigation of HAB spatiotemporal variability and interactions with aquaculture industries remained limited. Our study demonstrated that the remote sensing algorithm developed by Gokul et al. [17] provides a cost-effective and valuable tool for (1) producing an atlas of HABs to characterize their spatial patterns and seasonal variability in the Red Sea, and (2) investigating the linkages between HABs and the largest aquaculture facility in the region.

First, we discuss the possible oceanographic mechanisms that could potentially facilitate HAB events in the Red Sea. Our atlas reveals that the highest number of HAB events occurred in the SRS, followed by the SCRS and subsequently the NCRS, over the period of 2003–2017. The SRS and the substantial parts of SCRS are subject to a biannual influx of nutrient-rich waters from the Indian Ocean through the Gulf of Aden, driven by the seasonally reversing Indian monsoon winds [25,31–39]. During

the winter monsoon (October–April), prevailing south-easterly winds promote the northward advection of nutrient-rich surface waters into the Red Sea from the Gulf of Aden [32,40–42]. During the summer monsoon (May–September), prevailing winds reverse their direction, causing intense upwelling in the Gulf of Aden, ultimately generating an influx of nutrient-rich water through an intermediate water layer (Gulf of Aden Intermediate Water—GAIW) into the shallow waters of SRS [36]. This influx of nutrient-rich waters during both monsoons could disseminate HABs in parts of the SRS and SCRS regions throughout the year [12–14,17,18]. Besides the influx of nutrient-rich waters from the Indian Ocean into the Red Sea, the discharge of large quantities of nutrients from domestic, industrial, and agricultural wastes may also facilitate the growth of HAB species in the SRS and SCRS regions [8,9,11,18,20,21]. For instance, Alkershi and Menon [9] documented a dinoflagellate HAB event along the coastal waters of Yemen during March 2009. The high density of this dinoflagellate bloom (up to $5.5 \times 10^5$ cells $L^{-1}$) was observed at a station where large amounts of remains of slaughtered fishes and waste from fishing boats are discharged. We also note that HAB events in the Red Sea region were found to decrease northward and no events were detected by the remote sensing algorithm in the NRS over the study period. In addition, previous studies demonstrated that HABs in the NRS were often observed at low cell densities ($<1 \times 10^4$ cells $L^{-1}$), corresponding to low *Chl-a* concentrations ($<1$ mg $m^{-3}$) [18,43–45]. For instance, Madkour et al. [44] reported a cyanobacteria bloom (up to $7 \times 10^3$ cells $L^{-1}$) that was associated with *Chl-a* concentrations up to 0.9 mg $m^{-3}$, which also co-occurred with diatoms along the Egyptian coast of the NRS between 2007–2008. However, the remote sensing algorithm used in this analysis is only capable of detecting the presence of a HAB event when *Chl-a* concentrations and cell abundances exceed a minimum value of ~1 mg $m^{-3}$ and ~$1 \times 10^4$ cells $L^{-1}$, respectively [17]. We thus note that the HAB events were not detected by the remote sensing algorithm in the NRS region over the 15-year period from 2003 to 2017. A more detailed discussion of the possible oceanographic mechanisms that were responsible for the majority of HAB outbreaks in the three provinces of the Red Sea is narrated below.

In the NCRS, HABs were more frequent on the south-eastern coast of the region and some of these events were found to be consistent with the concurrent in situ sampling locations. For instance, the presence of a HAB event detected by the remote sensing algorithm is in good agreement with the study of Catania et al. [16], who documented the occurrence of the benthic dinoflagellate *Ostreopsis* sp. over the region of Thuwal (Saudi Arabia) during the summer of 2013 (Figure S1). Catania et al. [16] also revealed that HAB events occurred off the coast of Thuwal (Saudi Arabia) during the summer monsoon, when warm temperatures caused corals to bleach on the inshore reefs. The corals on these inshore reefs appeared to suffer high mortality rates and the increased availability of dead coral surfaces could have favored the proliferation of HABs in this region [16]. In addition, we detected a HAB event that was found to concur with Touliabah et al. [46], who reported a cyanobacteria bloom event in the south-eastern coast of NCRS (Jeddah) during the summer of 2004 (Figure S1). This bloom event was linked to the nutrient-rich domestic wastewater discharge along the coastal waters of Jeddah [46]. Our results also suggest a summer peak (in particular during May) in the time series of satellite-derived *Chl-a* concentration (a proxy for high nutrient concentration) during 2004 over the region of Jeddah, where the HAB event was detected (Figure S2).

In the SCRS, we observed the highest number of HAB events along the eastern coast of the region, and the majority of those events occurred during summer. This is also concurrent with several previous studies [8,10,11,18,20] that reported HAB events along the east coast of SCRS in the coastal waters of Al-Lith, Doga, and Jizan during summer. The remotely sensed HAB events were found to coincide with in situ observations acquired during previous campaigns [8,11] in Al-Lith and Doga during the summers of 2010 and 2004, respectively (Figure S1). The prevailing north-westerly wind patterns observed during summer 2010 (Al-Lith) and 2004 (Doga) (see Figure S1) may transport HABs towards the south along the east coast of SCRS [8]. While HABs were detected over large distances in the east coastal waters of SCRS, they mainly occurred locally, probably as a result of anthropogenic nutrients and waste produced from Al-Lith's aquaculture facility [8,10,11,18,20]. Kurten et al. [20] suggested

that the supply of dissolved and particulate organic matter from aquaculture effluents could lead to excessive nutrient loading that may disseminate HABs along the east coast of the SCRS during the summer monsoon. The time series of *Chl-a* concentration acquired in Al-Lith and Doga (east-coast of SCRS) during 2010, and 2004, respectively revealed that elevated *Chl-a* values (>1 mg m$^{-3}$; a proxy for high nutrient concentration) were observed over summer season (see Figure S2). Few studies have been conducted on the distribution and abundance of HABs in the western coastline of SCRS (in particular Sudanese Red Sea coasts) [47,48]. For example, El Hag and Nasir [47] recorded high cell densities (up to $1 \times 10^5$ cells L$^{-1}$) of harmful dinoflagellates along the coastal waters of Sudan. This high accumulation of HABs along the Sudan coastal waters was associated with the high nutrient loadings from the economical and industrial developments on the Sudanese Red Sea coast [49,50]. Previous studies investigated the circulation dynamics, including surface currents and mesoscale eddies, which may increase the availability of nutrients and occupy large areas in the open waters of the SCRS region [30,51]. These high nutrients could be responsible for the large-scale HAB events in the western coastline and open waters of the SCRS region [18,47,48].

The SRS has been subjected to more HAB events than anywhere else in the Red Sea, showing the most persistent and large-scale events in comparison to the SCRS and NCRS. Previous research has also observed HABs in the SRS (in particular the coastal waters of Yemen) over the last two decades [9,12–14,17]. We detected the presence of a HAB event in the coastal waters of Al-Hodeidah during winter 2009, which is consistent with the results of Alkershi and Menon [9] who documented a bloom of *N. scintillans/miliaris* during the winter 2009 in the coastal waters of Al-Hodeidah (Yemen) [see Figure S1]. Large quantities of nutrients were observed in the coastal waters of Al-Hodeidah after the remains of slaughtered fishes and waste from fishing boats were discharged in March 2009 [9]. The time series analysis over Al-Hodiedah (Yemeni coastal waters) also confirmed that elevated *Chl-a* values (>1 mg m$^{-3}$; a proxy for high nutrient concentration) were observed between February–March in 2009 (see Figure S2). Furthermore, south-easterly winds were clearly observed during winter 2009 over the SRS, which could have re-distributed HAB species to the open waters of the region (Figure S1). To support this observation, Gokul et al. [17] suggested that regional dynamic circulation could transfer HABs hundreds of kilometers away. In addition, our results are consistent with Alkwari et al. [14], who reported the occurrence of intense *P. bahamense* blooms (<$1\times10^5$ cells L$^{-1}$) along the coastal waters of Yemen during winter 2013 (see Station 'S2' in the HAB map of Al-Hodeidah region (winter 2013) in Figure S1). Besides, Alkawari et al., [14] reported a cyanobacteria *T. erythraeum* blooms that were observed as spatially less dense (>$1\times10^4$ cells L$^{-1}$) along with the dinoflagellates *P. bahamense* in the coastal waters of Yemen during winter 2013. These HAB events were linked to south-easterly wind patterns that promotes the northward advection of nutrient-rich surface waters into the Red Sea from the Gulf of Aden during the winter monsoon [14,18]. However, the remote sensing algorithm of Gokul et al. [17] is limited at detecting the cyanobacteria *T. erythraeum* along the coastal waters of Yemen during winter 2013 (see Station "S1" in the HAB map of Al-Hodeidah region (winter 2013) in Figure S1). We therefore acknowledge an uncertainty in the accuracy of the remote sensing model at detecting HABs that are spatially less dense, and/or composed of mixed assemblages of phytoplankton (especially cyanobacteria with dinoflagellates). We also note that the persistent presence of clouds, sun-glint, and sensor saturation over sand severely reduces the retrieval of satellite ocean color observations from the MODIS-Aqua data in the southern part of the Red Sea region (including SCRS) during the summer monsoon (June–September) [23–25,52,53]. This may inhibit the investigation of HABs in the region during the aforementioned period and thus could explain why we observed relatively few events in the SRS during summer (see Figure 3b). The satellite observations that are affected by the adverse atmospheric condition (Haze and clouds) may cause an uncertainty in determining the duration of HAB events over the summer monsoon. For example, Mohamed and Al-Shehri [11] reported the occurrence of a HAB event (raphidophyte *H. akashiwo*) between mid-May and mid-June (24–32 days) of 2010 in the vicinity of Al-Lith's aquaculture facility. However, these adverse atmospheric conditions may hinder the ability of the remote sensing algorithm of Gokul et al. [17] to detect the aforementioned

HAB event during June 2010. We therefore note that the duration of remotely sensed HAB events was found to be considerably lower in the Al-Lith region (9–16 days) during 2010, in comparison to the in situ observations.

　　HABs occur naturally in the Red Sea but may also be exacerbated with anthropogenically-driven activities such as pollution, local aquaculture expansion and ballast water transport [8–11,18,20,21]. The NAQUA is the largest aquaculture facility in the Red Sea, is an important contributor to the socio-economic development of the Kingdom of Saudi Arabia and helps to meet the growing demand of seafood products in the nation [21,54–57]. Although NAQUA is important for food and economic prosperity, it is a point source of anthropogenic nutrients to the coastal waters of Al-Lith [10,11,18,21,56]. A recent study suggested that the highest discharge of nutrients and waste follows the flushing of shrimp aquaculture ponds in the coastal waters of Al-Lith [21]. The aquaculture facility aims for three harvests per year and after each harvest, the ponds are flushed and dried for one month. The exact timing of flushing is unknown, although the harvest cycles suggest that the flushing of the aquaculture ponds takes place all year round [21]. Furthermore, the wastewater discharged from the aquaculture outfall may facilitate the proliferation of HAB species in an otherwise oligotrophic region during the stratified conditions in summer [10,11,20,21,30]. We found that the spatial coverage of HABs in regions nearby to NAQUA increased in conjunction with the average annual aquaculture production during the nine-year period (2002–2010) (Figure 4b). In addition, we noticed that satellite-derived *Chl-a* concentrations were also high (>1 mg m$^{-3}$) during the summer season and exhibited an increasing trend over the period of 2002–2010 (Figure 4c). This is analogous with the results of Hozumi et al. [21], who revealed that water discharged from the aquaculture facility was characterized by high in situ *Chl-a* concentration (>1 mg m$^{-3}$) during the summer season between 2014 and 2015. Thus, our results imply that HAB events have expanded both in time and space alongside intensified aquaculture production in the coastal regions of Al-Lith. Mohamed and Al-Shehri [11] also suggested that the intensive aquaculture operations of the NAQUA facility causes self-pollution as a result of excessive feeding and fish feces, causing eutrophication in the aquaculture area. This may disseminate HABs in the coastal waters of Al-Lith over summer monsoon. These HAB events may pose a serious threat to the sustainable exploitation of the aquaculture sector [11,21]. Thus, the expansion and sustainability of Al-Lith's aquaculture facility are dependent on the adoption of management practices that minimize the impact of excessive nutrient loading. For instance, as a part of the sustainability programme, Al-Lith's NAQUA facility is developing mangroves plantations in order to mitigate the environmental issues including impacts on public health, fisheries, and aquaculture (https://thefishsite.com/articles/sustainable-antibiotic-free-aquaculture-on-the-desert-coast). However, further efforts should be made in managing the aquaculture wastewaters, since there is a clear association between the excessive nutrient loading from aquacultures and increased phytoplankton biomass (including harmful algae).

　　From a management perspective, HABs in the Red Sea are considered to be one of the most important direct drivers of biodiversity loss and change in ecosystem services regionally [13,16,18]. As we have shown, our atlas can identify high-risk areas in the Red Sea. For instance, we have detected a potential high-risk area in the NCRS (particularly along the southeastern coastline). As this region supports an important level of biodiversity and may consequently sustain numerous communities of organisms throughout the Red Sea [38,58,59], we suggest that the NCRS may possibly require HAB management. In comparison to the NCRS, the SCRS has experienced a higher occurrence of HABs along its' eastern coast over the last two decades. This region is well known for its annual whale shark aggregations [21,60,61], yet shows an accumulation of HAB threats. Thus, there is a pressing need to conduct further research in this region in order to elucidate the potential detrimental impacts of HABs on marine organisms. We also highlight the possible need for HAB management in Red Sea regions other than the southern part, such as the central part of the basin, as these regions are subject to the substantial elevation of nutrient loading associated with the coastal urbanization, industrial expansion and aquaculture development along the Saudi Arabian Red Sea coast [8,10,11,18,20,46]. We also propose the SRS to be a prime candidate, as it exhibits the highest number of HABs as well

as the most persistent and large-scale events in the entire Red Sea. In addition, it is also a region where the seasonal inflow of surface water from the Gulf of Aden appears to play an important role in the maintenance of pelagic organisms and their distribution within the Red Sea [12–14,30,39,62,63]. Previous studies have also identified the region as the most highly HAB-affected part of the Red Sea and is often associated with severe fish mortalities and numerous ecological impacts [9,12–14,17].

## 5. Conclusions

Our results revealed that the highest number of HAB events occurred in the SRS, followed by the SCRS and the NCRS. In addition, HAB events detected in the SRS and SCRS were characterized by a longer duration and an increased spatial extent, in comparison to the NCRS. In particular, regions along the east coast (excluding the NRS) appear to be more at high-risk for the occurrence of HABs. HAB events exhibit a distinct seasonality, with a higher number of events occurring during summer in comparison to winter and autumn. Investigations of the potential linkages of HABs with the largest aquaculture facility in the Red Sea revealed that HABs have expanded, both in space and time, over the east coast of the SCRS in conjunction with intensified local aquaculture production. It is therefore possible that HAB events in this region have increased alongside farming operations, which could pose a serious threat to the sustainable exploitation of the aquaculture sector and the coastal environment. The implementation of integrated, conservation-oriented management plans are necessary if HABs become more frequent in the proximity of the aquaculture facility. Since the aquaculture industry is a financial asset for the Kingdom of Saudi Arabia, a national wastewater management program should also be implemented, in order to prevent the excessive nutrient loading into the oligotrophic Red Sea ecosystem.

**Supplementary Materials:** The following are available online at http://www.mdpi.com/2072-4292/12/22/3695/s1, Figure S1: Remotely sensed HAB events and the wind speed (m/s) patterns over the regions where the in situ sampling was conducted in the Red Sea during the different periods from 2003 to 2017, Figure S2: Time series analysis of satellite-derived Chl-a over the regions where the in situ sampling was conducted in the Red Sea during the different periods from 2003 to 2017, Figure S3: Remotely sensing HAB event in the Al-lith aquaculture facility, Figure S4: Remotely sensing HAB event in the Al-lith aquaculture facility, Table S1: Summary of the harmful algal bloom (HAB) species reported by the various studies in the Red Sea waters.

**Author Contributions:** Conceptualization: E.A.G., D.E.R. and I.H.; methodology: E.A.G. and D.E.R.; software: E.A.G.; validation: E.A.G.; formal analysis: E.A.G. and D.E.R.; writing—original draft preparation: E.A.G.; writing—review and editing: E.A.G., D.E.R., J.A.G. and I.H.; data curation: E.A.G.; supervision, D.E.R. and I.H.; project administration: I.H.; funding acquisition: I.H. All authors have read and agreed to the published version of the manuscript.

**Funding:** This research was funded by the King Abdullah University of Science and Technology (KAUST) Office of sponsored Research (OSR) under the Virtual Red Sea Initiative (Grant # REP/1/3268-01-01).

**Acknowledgments:** The authors are grateful to the Ocean Biology Processing Group of NASA for the distribution of the MODIS-Aqua and MERIS datasets and the development and support of the SeaDAS software. The authors are also grateful to the ESA Ocean Colour CCI team for providing and processing the *Chl-a* dataset. We are also grateful to the Sea Around Us project for providing the annual aquaculture production dataset.

**Conflicts of Interest:** The authors declare no conflict of interest.

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
