# Peer review of "Developing an Atlas of Harmful Algal Blooms in the Red Sea: Linkages to Local Aquaculture"

_remotesensing, doi:10.3390/rs12223695_

Round 1

Reviewer 1 Report

In my opinion, the manuscript can be published as is.

The authors draw a conclusion that the functioning of aquaculture is related to the frequency of planktonic blooms. They proved that when starting aquaculture, additional measures are needed to reduce the share of these investments in sea eutrophication. Authors used the descriptive method. I must say that it was not easy for me to combine all the information and consider that the conclusions are justified. Personally, I would prefer if the conclusions would result from the statistical analysis of observational results. 

However, I must say that the manuscript is written carefully, though a bit lengthy.   

Reviewer 2 Report

I would like for you contribution. The article is very interesting.

Reviewer 3 Report

  1. Summary

The authors clearly identified an issue (complete understanding of HABs) that seems to be prescient for the Red Sea while also being relevant to other areas of the world. The approach was fairly basic (merely applying algorithms developed by others, not really extending them, critically analyzing, presenting any idea of how uncertain estimates are, etc.), and the results/discussion were also relatively basic (summarizing counts, duration, and area of HAB events). The authors describe two main research activities used to address the issue: the creation of an atlas (spatial history of chlorophyll) and a case study. Some connections are made to historical local aquaculture activities, and other contributing factors (e.g. wind patterns) were discussed in a limited way.

Multiple remote sensing-derived products were used in the analysis, none of which were novel to this study. This paper provides a somewhat shallow presentation of the application of these products, not providing much context (e.g. The model for HABs could have been described in a bit more detail (even if development and validation were the subject of another paper). There was little indication in the manuscript that the supplementary material provided any evaluation/validation of the model, and what is included there simply shows that there were instances of agreeance. How often was the model incorrect? Additionally, for the chlorophyll model, how does uncertainty of its estimates compare to the range of observations in these regions of the Red Sea? Some level of certainty/accuracy needs to be presented along with the application.

This paper would greatly benefit from digging deeper into the mechanisms (i.e. providing a more detailed, evidence-based, quantitative analysis of how these mechanisms are observed. Additionally, a stronger analysis of the remotely-sensed products would make the findings of the paper more meaningful.

  1. Major Comments

The discussion of what might be contributing to blooms was really narrative and anecdotal (see also comments about section 3.2). It would have made a much stronger case to actually provide data alongside the patterns observed from the atlas instead of just describing these phenomena and where they take place (e.g. time series of wind data or nutrient concentrations at points could be provided to demonstrate localized effects of these factors). I would suggest that some of the discussion section could be moved to the results section if the exploration of mechanisms was actually backed up by data instead of just described in broad terms.

Is the scale in Figure 5 accurate? Can the algorithm detect the difference between 0.76 and 1.51 mg/m-3 accurately? Moreover, does a chl concentration of 3 mg/m-3 constitute a bloom? This seems like incredibly small variations of chlorophyll, especially given the margin of error that can come simply from diurnal variability or differences in sampling methods.

  1. Minor Comments

Quality of all figures should be improved – the text was blurry, making it more difficult to read without zooming in.

49: The authors do well to point to previous studies of HABs in the Red Sea. However, it’s not really clear what those studies are lacking. It would help to have a more descriptive or explicit review of what the previous studies covered/how they were limited.

Section 3.2: The delineation of the region for Al-Lith was not clearly defined. Why is the area limited to that box? The designation of that region could have an effect on the analysis; for example, there are stats provided for mean chl-a, but it isn’t clear how large that area is, or whether that area is representative of the NAQUA activities (and not influenced by other activities in the surrounding coastal area). Is there any evidence provided to indicate why the hydrodynamic influence/interactions of the Red Sea and the NAQUA facility are limited to that area?

The “case study” was really insufficient – looking at one image and noting that there are higher concentrations near a point during that one image is really just anecdotal. This feels like a very superfluous addition.

Throughout 3.1.1-3.1.3 and Figure 2: Is it fair to say duration is 8, 16, or 24 days? At 8-day return period, a bloom that lasted 15 days, would be lumped in with those that only lasted 1 or 2 days (but happened to occur on the same date as a satellite. No discussion of how some images might have been affected by cloud cover, affecting usability and causing further errors in duration.

156: When describing the results, is it appropriate to say that there were “up to 3 events”? Wouldn’t it be at least 3 events (it could have been more if the image acquisition dates weren’t limited)?

268: Wording: water masses ?

378: higher compared to what? To other areas? Compared to previous decades? Higher concentrations compared to some water quality standards? Simply because they are increasing does not adequately tell why it is an issue worth addressing.

394: The authors say that the “atlas presented in this study may serve as a valuable tool,” yet they did not make the atlas available or indicate how the products (time-series of maps) could be accessed. Some thought should be given to how make the results of the analysis into usable tools.

Author Response

The detailed comments of the reviewers are presented in non-bold text and our reply is presented in bold text. We have also attached a separate word file named " Response to Reviewer_3 comments" for your convenience.

Reviewer 4 Report

The manuscript aims to identify the long-term spatial and temporal distributions of harmful algal blooms (HABs) in coastal waters of the Red Sea using remote sensing and to investigate a potential linkage of increase in HABs occurrence with the aquaculture production. The reviewer appreciates authors’ every effort drawn in the manuscript. Overall comments are as follows.

  1. There is a discrepancy of data availability in between 15-year spatio-temporal distribution of HABs and 9-year interaction of HABs with aquaculture. The result showed that relatively high HABs events have been observed in SRS and SCRS, compared to Northern regions. We could understand the distribution of HABs from the 15-year remotely sensed data. But this study only analyzed interaction of HABs with local aquaculture production using 9-year data from 2002 to 2010. How do we understand their interactions from 2010 to 2017? Is aquaculture production still an essential factor for the intensification of HABs occurrence in the region?
  2. This study focused on frequency of occurrence, spatial coverage, and periods of HABs event. What kind of algal species have been usually observed as HABs? It can be helpful to understand the HABs event in the Red Sea, if there is a description about the species that have been observed.

Minor points

- Line 62, what is meaning of “atlas” in this study?

- Line 144 to 145, “less frequent events occurring during the winter and autumn (Figure 3).”
Based on the figure, the frequency of event looks like similar between summer and winter but the region where shows high frequency is different between them. Do you mean the comparison of total frequency?

Author Response

The detailed comments of the reviewers are presented in non-bold text and our reply is presented in bold text. We have also attached a separate word file named " Response to Reviewer_4 comments" for your convenience.

Round 2

Reviewer 4 Report

The reviwer thinks that the revised manuscript is enough to be accepted for publication without any further changes.